# Transfer Learning and Analogical Inference: A Critical Comparison of Algorithms, Methods, and Applications

Kara Combs [1] , Hongjing Lu [2,*] and Trevor J. Bihl [1]

1    Sensors Directorate, Air Force Research Laboratory, Wright-Patterson Air Force Base, Dayton, OH 45433, USA
2    Departments of Psychology and Statistics, University of California, Los Angeles, CA 90095, USA
*    Correspondence: hongjing@g.ucla.edu

**Abstract:** Artificial intelligence and machine learning (AI/ML) research has aimed to achieve human-level performance in tasks that require understanding and decision making. Although major advances have been made, AI systems still struggle to achieve adaptive learning for generalization. One of the main approaches to generalization in ML is transfer learning, where previously learned knowledge is utilized to solve problems in a different, but related, domain. Another approach, pursued by cognitive scientists for several decades, has investigated the role of analogical reasoning in comparisons aimed at understanding human generalization ability. Analogical reasoning has yielded rich empirical findings and general theoretical principles underlying human analogical inference and generalization across distinctively different domains. Though seemingly similar, there are fundamental differences between the two approaches. To clarify differences and similarities, we review transfer learning algorithms, methods, and applications in comparison with work based on analogical inference. Transfer learning focuses on exploring feature spaces shared across domains through data vectorization while analogical inferences focus on identifying relational structure shared across domains via comparisons. Rather than treating these two learning approaches as synonymous or as independent and mutually irrelevant fields, a better understanding of how they are interconnected can guide a multidisciplinary synthesis of the two approaches.

**Keywords:** transfer learning; analogical reasoning; generalization; artificial intelligence; cognitive science; analogical inference; learning; inference

## 1. An Introduction to Human and Artificial Intelligence Learning

Learning is a fundamental ability for any intelligent system, allowing it to gain knowledge, understanding, or skill through experience, active study, or guided instruction. Humans are in a state of constant learning through our observations, interactions, and experiences, from which schemas (abstract cognitive structures for organizing knowledge) are constructed [1–3]. Many research directions in cognitive science, neuroscience, and psychology focus on learning [3,4]. Research suggests that in humans, learning often involves the generation of relations between individual concepts [1,5,6]. Relational information links new information to that previously stored within a schema and guides the adjustment of current schemas or the creation of new ones [2].

However, before the creation of initial schemas and relations, one must have interaction(s) with the new or unfamiliar concept(s) [2]. These interactions are initiated by the senses, followed by the retention of information through one's short-term and/or long-term memory [7]. Depending on the task at hand, learning may require a significant amount of experience, practice, and interactions (humans' version of "training data"). However, for some tasks, humans can learn from very little experience [6,8]. Much of the work in artificial intelligence (AI) aims to develop highly accurate algorithms trained on large amounts of data and/or improve the amount of time needed to train. There is also a push to make AI "adaptive," such that it can easily adjust to changes in the environment based on its

perception or action/behaviors. However, learning via inference can offer a more efficient, less computationally expensive approach. The active field of machine learning (ML) often seeks to mimic human-based learning (see [3,9,10]). Before discussing computer-based implementations that leverage human abilities in inferencing, schema construction, and relationship generation, a broad understanding of machine learning is needed. This lays the foundation for the major aim of the paper, which is to compare two types of ML: transfer learning and analogical inference. As both methods' primary mechanism is the "transfer" of knowledge from a source domain to a similar target domain, many may see the two methods as equivalent since the differences have not been explicitly stated in the literature. Through this article, we aim to formally define the two approaches.

This paper provides a high-level background on machine learning in Section 2, transfer learning in Section 3, and analogical inference (which we use to encompass all analogical-based processes, including reasoning, learning, and mapping) in Section 4. Discussions of algorithms, methods, and applications that utilize transfer learning, analogical inference, or both are presented in Section 5, specifically including computer vision applications (Section 5.1), natural language processing applications (Section 5.2), and a summary (Section 5.3). Section 6 provides suggested directions for future work and finally, the conclusions are presented in Section 7. This review paper aims to:

1.  Identify critical differences and similarities between transfer learning and analogical inference;
2.  Review relevant evidence from the fields of computer vision and natural language processing;
3.  Make recommendations for future research integrating transfer learning and analogical inference.

## 2. Brief Overview of Machine Learning

Within ML, one can largely divide the field through families of methods, such as "symbolist," "connectionist," "Bayesian," "evolutionaries," and "analogizers," as summarized in Table 1 [11,12]. Notably, while "analogizers" semantically appear to be related to "analogical inference," this is only tangential, as the methods in this space refer to spatial similarity of classes based on distances between groupings in a data space [11]. Alternatively, ML algorithms can also be described in terms of different paradigms as it pertains to training, such as "supervised," "unsupervised," or "semi-supervised" [13]. Thus, to understand how these are related but differ from the topics of interest—transfer learning and analogical inference—there is a need to formally establish the major differences between common types of machine learning, as shown in Figure 1.

**Table 1.** Summary of Domingos's ML algorithm family definitions (see [11] for original).

| Family of Learning | Description |
| --- | --- |
| Symbolist (AKA inductive) | An inductive-reasoning-based method which is based on representing concepts through "symbols" which can be mathematically manipulated |
| Connectionism | A bottom-up approach inspired by the brain's ability to function based on firing neurons ("activation") |
| Bayesian | A technique based using on estimated event probability to predict outcomes, popular in causal situations |
| Evolutionary | A biological-inspired approach based on the ability to evolve and mutate in hopes of achieving better performance |
| Analogizers | A method based on identifying patterns and similarities between two domains and learning through various types of reasoning |

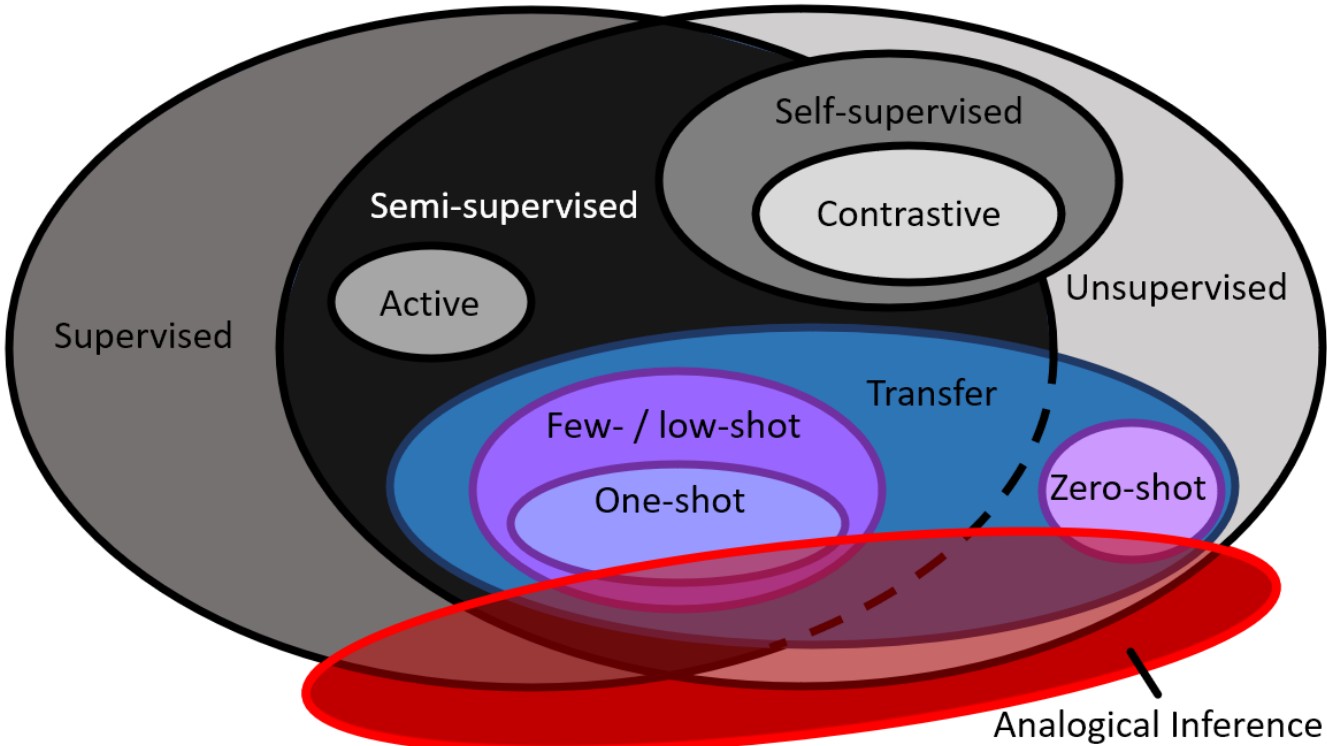

**Figure 1.** Venn diagram of popular machine learning algorithms: Hierarchy and relationship(s) between common types of machine learning are presented along with the two types on which this article focuses, transfer learning and analogical inference, which are denoted within the blue and red regions, respectively. Purple regions represent types of learning that often fall under either category. Set and set-overlap sizes are not necessarily depicted to scale; emphasis is on the general overlap between sets.

All ML applications can be split into three categories: supervised, unsupervised, or semi-supervised learning. Supervised learning (large dark-grey section on the leftmost side of Figure 1) involves the prediction of a classification label or value (in the case of regression) where the data (used for training and testing) are all labeled/known during the evaluation [13,14]. On the other hand, unsupervised learning (light-grey section on the rightmost side of Figure 1) only utilizes unlabeled data, typically for clustering or association problems [13,15]. Sometimes, both supervised and unsupervised learning methods can be utilized in a hybrid approach between the two where some of the data are labeled and some are not labeled; this is called semi-supervised learning (black section of Figure 1) [13,14,16].

In addition to the three broadest types of learning, other specialized learning algorithms have been developed in the past decade. Self-supervised learning is a subset of unsupervised or semi-supervised methods where the algorithm creates "pseudo-labels" to mimic and be solved as if it was a supervised problem [17]. Within self-supervised learning algorithms, contrastive learning is a popular approach in which pseudo-labels are created by comparing the data samples to one another to capture the similarity between samples. The learned feature representations then impact the results of downstream tasks, such as classification/clustering of the data [17,18]. Most recently, contrastive learning has included significant efforts involving Generative Adversarial Networks (GANs). In the computer vision realm, GAN methods consider the curation and expansion of datasets when limited data are available for training [19]. Depending on how much original data are labeled, if any, self-supervised and contrastive learning algorithms fall either in the greater domains of semi-supervised or unsupervised learning. A type of learning entirely within the semi-supervised is active learning, where an "oracle," commonly a human, is

used to derive information about or assess the algorithm and/or its performance, such that the feedback can be incorporated to improve the algorithm [14,16].

Of primary interest, herein, transfer learning (blue oval in Figure 1) is defined as a technique where some portion of knowledge from one domain is transferred and applied to a different but related domain [15,20]. Like self-supervised learning, transfer learning may be supervised, semi-supervised, or completely unsupervised, depending on whether the data labels are known or not. Many transfer learning algorithms are classified into few-shot (also called low-shot), one-shot, and zero-shot. Few-shot learning is a semi-supervised approach that uses a very small dataset of labeled instances for training (or only one training instance in the case of one-shot) [20–22]. Zero-shot learning is a method where a pretrained algorithm applies its knowledge learned from its initial training data to completely "unknown" data that the algorithm has never been exposed to before [20,23,24]. Though the pretrained algorithm may be within the supervised or semi-supervised domains, its application to unlabeled data is key to zero-shot learning's classification within the unsupervised learning field.

The shot family of learning is one of the main bridges connecting transfer learning and analogical inference (red oval near the bottom of Figure 1). Analogical inference is a method in which an analogy is formed between two different but related domains to draw inferences about the target domain (unfamiliar to algorithm) and its relationship with the source domain (familiar to algorithm) [25,26]. The vast majority of ML algorithms can be classified into the categories shown in Figure 1; however, these types can be combined with other types of learning, including, but not limited to: deep, reinforcement [13,14,20], ensemble [27], multi-task [14,15], multi-view [14,15], online [14], and many more that are beyond the scope of this article. A summary table of the types of machine learning is presented in Table 2.

The two types of ML on which this paper focuses, transfer learning and analogical inference, both involve the transfer of knowledge from one scenario and applying it to a different related scenario. Current work in AI has paid little attention to human analogical inference and learning; indeed, within the field of AI, the colloquial term "transfer learning" is often used without consideration for other potential labels due to a lack of awareness and popularity. However, there are fundamental differences between the two learning approaches, as discussed throughout the remainder of this paper. Work in cognitive science has yielded both rich empirical findings from human experiments and general theoretical principles underlying human analogical inference and learning. Similarly, new machine learning algorithms for transfer learning have led to important advances, but also face serious hurdles if they are to achieve truly adaptive learning. Rather than either treating these two learning approaches as if they were the same or as independent and mutually irrelevant fields, they share elements that can promote new advances [12]. A synthesis of the two approaches will benefit both the AI and cognitive science research communities by advancing our understanding of both human and machine intelligence. In the next two sections, we will describe transfer learning and analogical inference, and then review how they have been specifically utilized individually or together within the computer vision and natural language processing domains.

**Table 2.** Types of machine learning methods.

| Type of Learning | Description |
| --- | --- |
| Deep | A subset of ML that utilizes artificial neural networks (ANNs) with three or more layers, sometimes called "deep neural networks (DNNs)" |
| Supervised | Prediction of a classification or value (in the case of regression) where the data (used for training and testing) are all labeled and known |
| Unsupervised | Utilizes only unlabeled data typically for clustering or association problems |
| Semi-supervised | A hybrid approach between supervised and unsupervised involving labeled and unlabeled data |
| Self-supervised | A type of unsupervised and/or semi-supervised methods where arbitrary labels are created for each data instance allowing it to act as a supervised learning problem |
| Contrastive | Type of self-supervised methods where data instances are compared to one another to determine their clustering into classes |
| Reinforcement | Approach utilizing a reward/penalty system upon evaluation of the environment and its goal(s) |
| Multi-view | An instance where a specific problem can be accurately viewed in two different manners |
| Multi-task | Technique focused on solving multiple related problems/tasks simultaneously |
| Ensemble | A method that combines individual models into a larger model to improve performance |
| Active | A specific type of semi-supervised learning where an "oracle," commonly a human, is used to derive information about or assess the algorithm and/or its performance |
| Online | A method that is best applied when the data to train the algorithm is introduced sequentially as the algorithm's parameters are fine-tuned |
| Zero-shot | Type of learning where an algorithm has "unseen" data outside of its initial training dataset with the ability to accurately identify and classify said unseen data |
| Few-shot/low-shot/one-shot/multi-shot | A semi-supervised approach that uses a very small dataset of labeled instances (or only one instance in the case of one-shot) |
| Transfer | A technique where some portion of knowledge from one domain is transferred and applied to a different but related domain |
| Analogical inference | Method where an analogy is formed between two different but related domains to draw inferences between each other as well as specifically the target domain |

## 3. What Is Transfer Learning?

Transfer learning became an active research area in the early 1990s, though the technique and its applications date back to the mid-1970s [28]. Transfer learning differs from other traditional machine learning methods because of its focus on the interaction and "transfer" of knowledge from a source model to a target model. As shown in Figure 2a, a traditional approach would treat the source and target datasets completely separately and train a separate model for each. In contrast, a transfer learning approach (shown in Figure 2b) would create a model from the source dataset and utilize that knowledge to train a model for the target dataset.

There are several benefits to utilizing transfer learning over traditional methods. These benefits include the elimination of the following requirements/expectations of traditional ML methods:

1. Necessity of sufficiently large amounts of high-quality and labeled training (and testing) data in the target domain [15,29–31];
2. Requirement for the training and testing datasets to share the same feature spaces [31];
3. Need for significant human interaction (time and monetary compensation therein) for annotation, verification, etc., during the data curation stage and/or active-learning-like techniques [31].

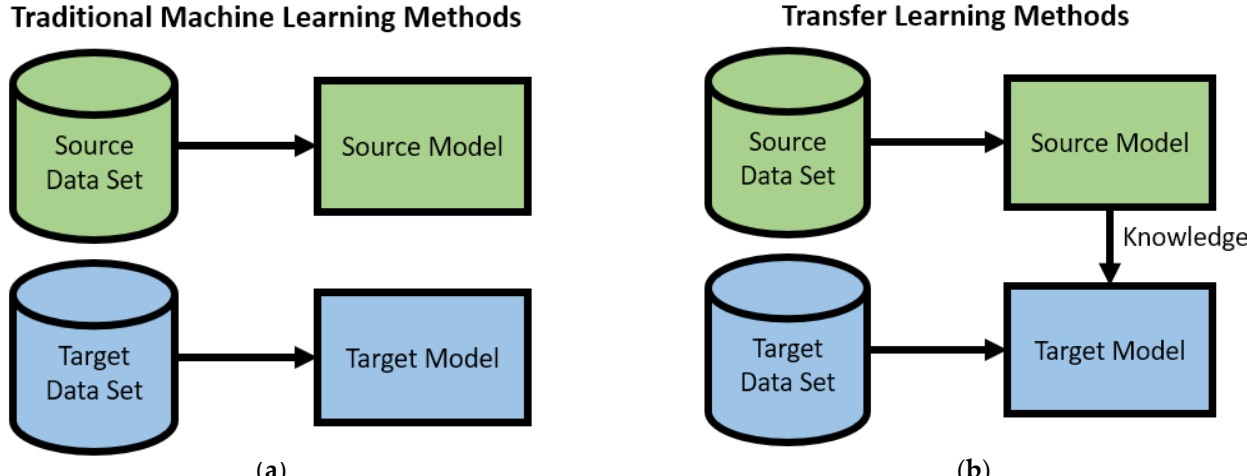

**Figure 2.** Difference between traditional and transfer ML methods: (**a**) source and target datasets are used to train separate models that do not interact; (**b**) source dataset is used to train a model and the derived knowledge is used to influence the target model for the target dataset.

However, transfer learning is not a "one-size-fits-all" technique due to its underlying assumptions. Any time the transfer of knowledge occurs, there are assumptions regarding how transferrable the knowledge is from the source to the target domain. The key assumptions are reasonable degrees of similarity between the source and target domains [31] and prior knowledge regarding what knowledge is to be transferred and when [20,31].

The goal is that the transfer is "positive," meaning the knowledge acquired from learning in a source domain improves the algorithm performance for the target domain, but there is always a risk of "negative" transfer where the source knowledge does not help and may even hurt performance in the target domain [15,30–32]. The probability of "negative" transfer is higher for "far transfer" (source and target are farther away semantically) problems compared to "near transfer" (source and target are closer semantically) [8]. Taking a simple example presented in [33], a near transfer analogy example would be "Blindness is to sight as deafness is to hearing" all of which occurs within the general domain of "human senses." Using the same source words, a far transfer analogy would be "Blindness is to sight as poverty is to money," where the former word pair is still from the domain of human senses, but the latter word pair is from the different domain of economics [33]. Despite the relation between the word pairs remaining constant, the far transfer analogy is often more difficult to infer in comparison to the near transfer analogy problem.

Figure 3 shows an example architecture of a transfer learning algorithm, which uses an artificial neural network (ANN) consisting of convolutional and dense (i.e., "fully connected" and/or "classification") layers. In this example, a source dataset, which is usually quite large, is used to train a model to make predictions using traditional methodology. When a target dataset is selected, instead of repeating this training process for the target dataset, the feature extraction elements of the source model (such as the number of layers and their weights) are utilized within the target model architecture. The layers and weights re-utilized from the source model are "frozen." Note that there are no expectations of how many layers or which types of layers are frozen. However, depending on whether the source and target dataset have different classes, the target model might need to add new dense layers that align with the different (if any) prediction classification categories. There can also be the addition of new layers; however, there are many combinations of frozen, unfrozen, and new layers that will yield different results.

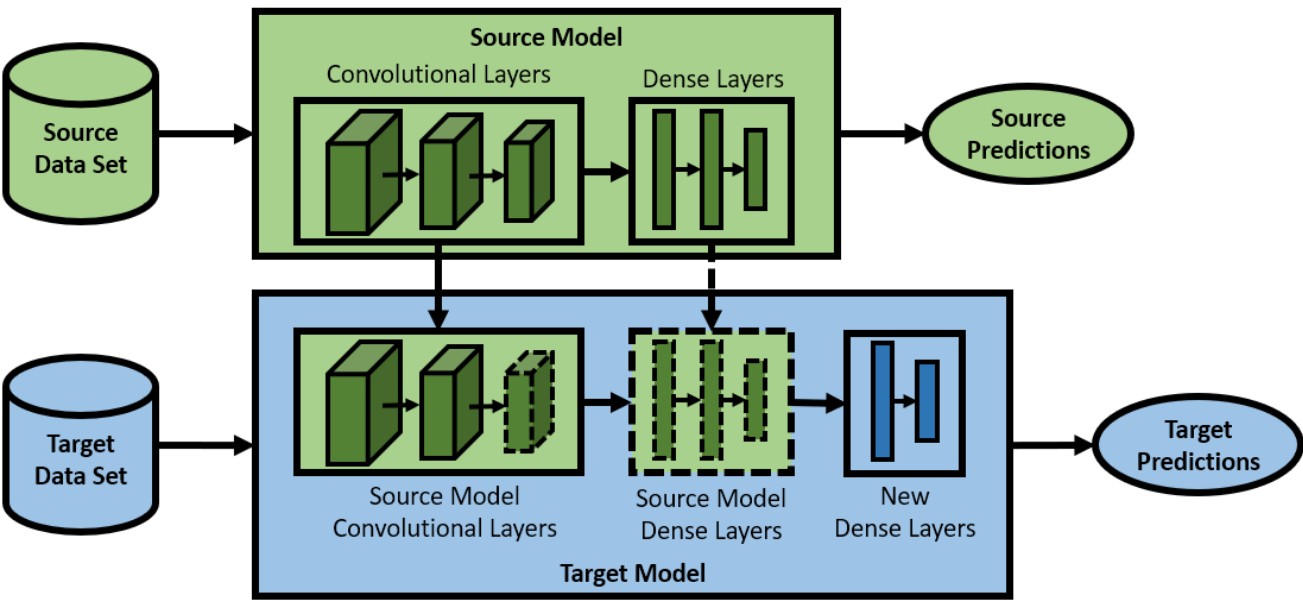

**Figure 3.** General transfer learning architecture; example optional elements have a dashed outline.

One question to be answered within every transfer learning problem is how to balance the ratio between frozen, unfrozen, and new layers. This is often derived from the size and similarity differences (if any) between the source and target datasets. Consider the various architectures of a theoretical ANN in Figure 4. Figure 4a represents the source ANN architecture, which, at its most basic level, consists of convolutional and dense layers. Regardless of the input dataset, the ANN in Figure 4a will always be agnostic and produce predictions based on the classes associated with the source dataset. Many applications will implement architectures as shown in Figure 4b,c, which involve allowing some dense layers to be unfrozen and/or added in the case of Figure 4c specifically. Though the convolutional layers of the model stay the same, the classification (dense) layer parameters can fluctuate. Convolutional layers can also be unfrozen (see Figure 4d) or added (see Figure 4e). Finally, for completeness, if the entire architecture is new, the model is completely different and no longer deemed transfer learning. As one allows more layers to be unfrozen or adds new layers, the resulting model adapts more to the target dataset.

At its instantiation, data are considered to be "raw" in the sense that they are typically in need of some pre-processing (value validation, removal of duplicates, standardization, etc.) and considered to be highly dimensional (instances are described by many variables) [34,35]. Dimensionality reduction (assumed to encompass feature selection (see [36–38]) and extraction (see [39])) are often used to represent the "raw" data using "features," [39,40]. These features are often vectors, which can be represented (and sometimes visualized) within a "feature space." This is an important step of representation to cope with computational resource, mathematical feasibility, and algorithm complexity when many dimensions are involved [39,40]. However, as these raw inputs are manipulated and turned into "features," they need to be represented in a new way because they have changed from their original "raw" state. These concepts of "feature spaces" are important when discussing transfer learning. For example, one of the most popular types of transfer learning is domain adaptation, which is where the feature spaces of the source and target domains remain the same [15,30]. A "domain shift" (or gap) exists when these feature spaces are different [41,42].

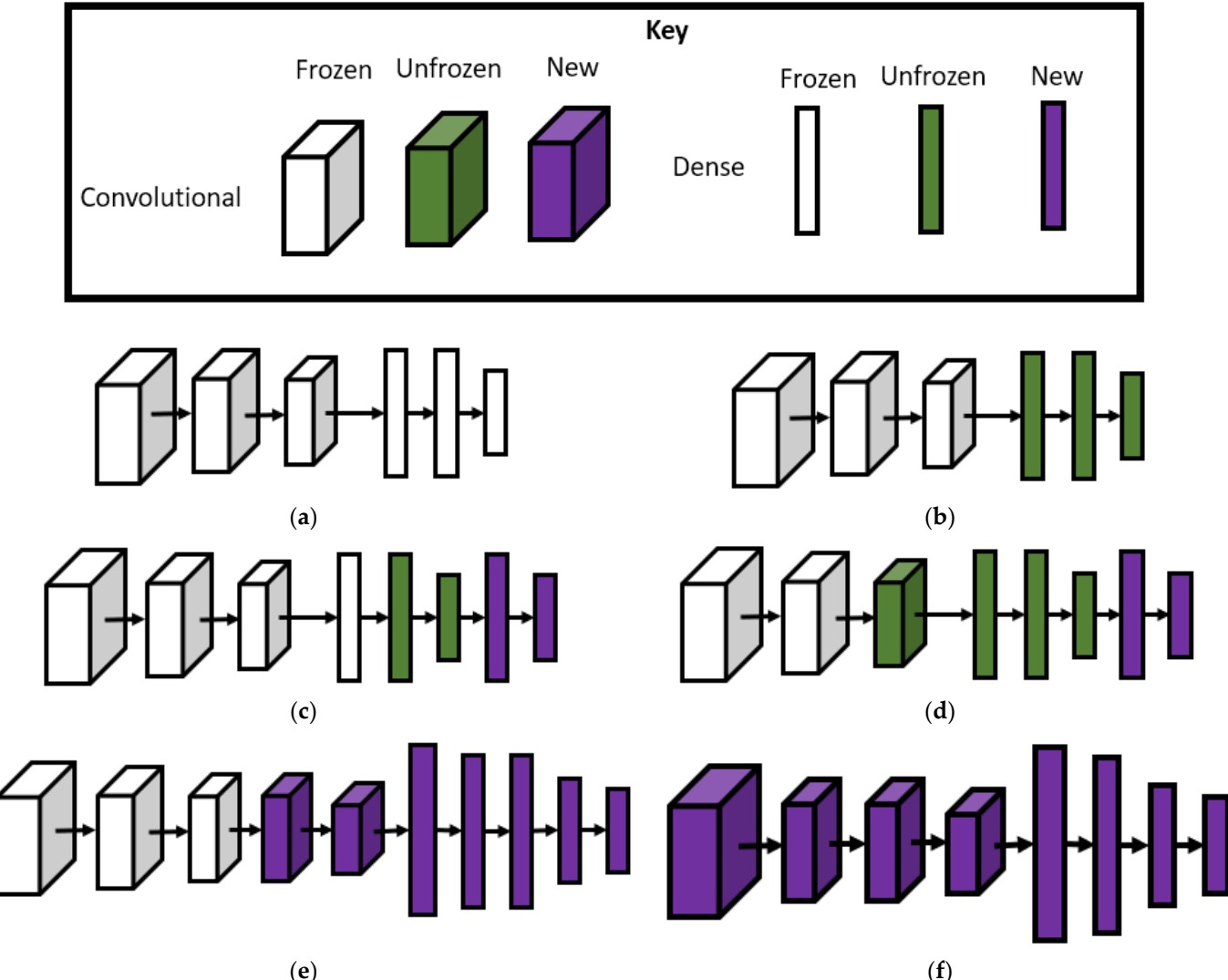

**Figure 4.** Varying amounts of "knowledge" transferred in a transfer learning application involving ANNs: (**a**) original source mode; (**b**) frozen convolutional layers with unfrozen dense layers; (**c**) frozen convolutional layers with a combination of frozen, unfrozen, and new dense layers; (**d**) combination of frozen and unfrozen convolutional layers and a mix of unfrozen and new dense layers; (**e**) combination of frozen and new convolutional layers with all new dense layers; (**f**) all new convolutional and dense layers, i.e., a completely separate model.

There are many ways to categorize transfer learning algorithms and methods. Some surveys classify transfer learning into two primary types depending on the relatedness between the source and target domains: homogenous (source and target domains share the same feature space) and heterogenous (source and target domains have different feature spaces) [41,43]. One study additionally included negative transfer learning, aiming to understand the association between the source and target domains and narrowing the gap in association (if any) through vector space manipulation [30]. Other surveys classify types based on the availability of data labels: inductive (only target domain has labels), transductive (only source domain has labels), or unsupervised (neither domain has labels) [31,44,45]. A more recent survey proposed two overlapping hierarchies, within which transfer learning algorithms can be viewed as data-based (focused on minimizing differences between the features of the source and target data) or model-based (focused on accurate predictions on the target data given the source data and model) approaches [15].

Transfer learning is a large and quickly growing domain; in contrast, analogical inference is a smaller field with steadier growth over time.

## 4. What Is Analogical Inference?

In contrast to the AI realm, in which transfer learning is popular, analogical learning has been a foundational pillar in cognitive science [25,26]. In its simplest terms, an analogy is a comparison between a familiar domain (called the "source") and an unfamiliar domain (called the "target") [25,26,46]. Analogical reasoning—the ability to find and exploit similarities based on relations among entities, rather than solely on the entities themselves—is a hallmark of human intelligence and creativity. Humans routinely use analogies in daily life. Consider, for example, this description of the difficulty in using a website: "The website is as dumb as a box of hammers and as useful as a paper teapot." Such analogical descriptions not only highlight core issues but also make the description more interesting and more likely to be perceived as creative.

Analogy problems are a common measure in human intelligence tests, as an analogy can be considered a prime example of zero-shot or few-shot learning—the ability to make inferences with minimal prior exposure to structurally similar problems, which is a general characteristic of human intelligence. One form of visual analogy problem is the Raven's Progressive Matrices (RPMs) [47], consisting of geometric shapes that undergo systematic transformation across the rows and columns of a 3x3 matrix. To emulate human reasoning ability, AI scientists have recently developed deep learning systems to solve RPM-style problems [48–50]. The deep learning approach to analogy is to view it as a task for which a deep neural network can be trained end to end by providing massive data consisting of reasoning problems. However, the success of these deep learning models, like other models in transfer learning, depends on high similarity between training problems and test problems and datasets of massive numbers of reasoning problems (e.g., 1.42 million problems in the PGM dataset [48] and 70,000 problems in the RAVEN dataset [49]).

This dependency on direct training in an analogy reasoning task using big data marks a stark contrast between the transfer learning approach and human analogical reasoning. When the analogy task is administered to a person, "training" is limited to general task instructions with, at most, one practice problem. Because the task is of interest as a measure of fluid intelligence—the ability to manipulate novel information in working memory—extensive pretraining on analogy problems is neither necessary nor desirable. In addition, human analogical reasoning can handle a wide range of complex inputs, such as pictures, movies, text, and daily experiences, rather than being constrained to specific feature spaces in transfer learning. An example situation is visualized in Figure 5. Though a seemingly unrelated and nonsensical comparison between a human body and a tree, human children can utilize analogical inference to estimate where a tree's theoretical "knee" would be.

Research on analogical inference includes both human studies (see [51–55]) and algorithm creation. Analogical inference has a deep history involving visual (see Figure 6a,b) and textual analogies (see Figure 6c–e). The first analogical inference algorithm was ANALOGY, which was designed to solve visual geometric analogy problems of the sort shown in Figure 6a [56]. This "A is to B as C is to D" (often abbreviated as *A:B::C:D*) format is also popular for verbal analogies, as shown in Figure 6c. These problems have several varieties, exemplified by standardized tests, such as the Scholastic Aptitude Test (SAT) or American College Testing (ACT), for high school students (see [20]). Briefly returning to the visual analogy scope, many modern algorithms focus on 3-by-3 matrix-like problems, as exemplified in Figure 6b. Figure 6a–c present the respondent(s) with options for the missing element, rather than requiring the respondent to generate their response (though this is not always the case for textual A:B::C:D problem, as in Figure 6c). There are problems for which the respondents are not presented with choice options. Examples of these are sentence-based analogies, such as the metaphor shown in Figure 6d and the story-based analogy in Figure 6e. These analogy problems were the initial starting point of the development of analogical inference methods (more commonly referred to as "analogical reasoning").

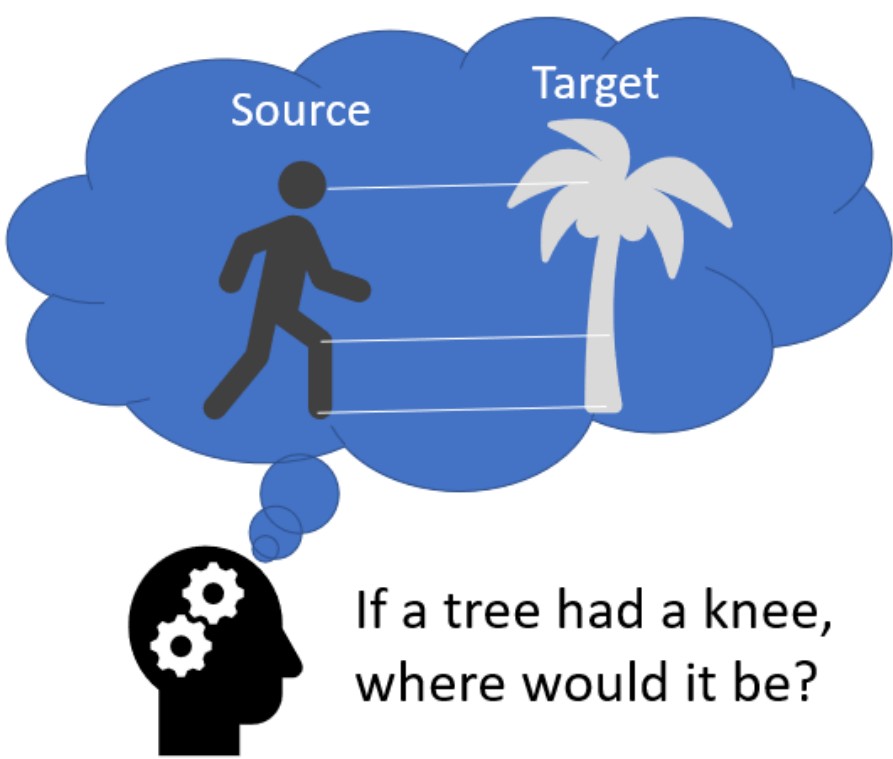

**Figure 5.** Visualization of how a human child can utilize analogical inference between two seemingly unrelated domains.

The majority of work in analogical inference has been conducted by cognitive scientists (see [57–59]). Regardless of whether an algorithm was created for verbal or visual reasoning, relation representations and inference are the key components in analogy. However, most cognitive-science-inspired algorithms for analogical inference can be classified into symbolist, connectionist, or hybrid families based on how the analogy is deconstructed and represented [59]. These families are elaborated on in Section 5.2. Schoonen argues that much is centered around the idea of "relevant similarity" [60]. This is reasonable since an analogy is most basically considered to be a comparison between two similar situations. However, as demonstrated in Figure 5, the source and target domains do not necessarily need to be "alike" in the sense that an exact, universally agreed-upon mapping exists. Analogical inference has been considered a successful approach to abstraction, which can be key when solving problems with similar structures in different domains, to generalize a situation or remove unneeded information [61]. Some research suggests that analogies are formed by recruiting specialized brain regions and networks to process relation information [62].

Despite being of various forms and structures in analogical tasks (see Figure 6), according to [25,26], previous research in cognitive science has identified three core processes in analogical inference, which are also included in computational algorithms of analogy:

1. Retrieval—accessing a similar scenario from long-term memory;
2. Mapping—the aligning of elements, structures, and concepts between the source and target;
3. Evaluation—judgment of the quality of specific aspects, such as the inferences made, mapping created, and/or the analogy in general.

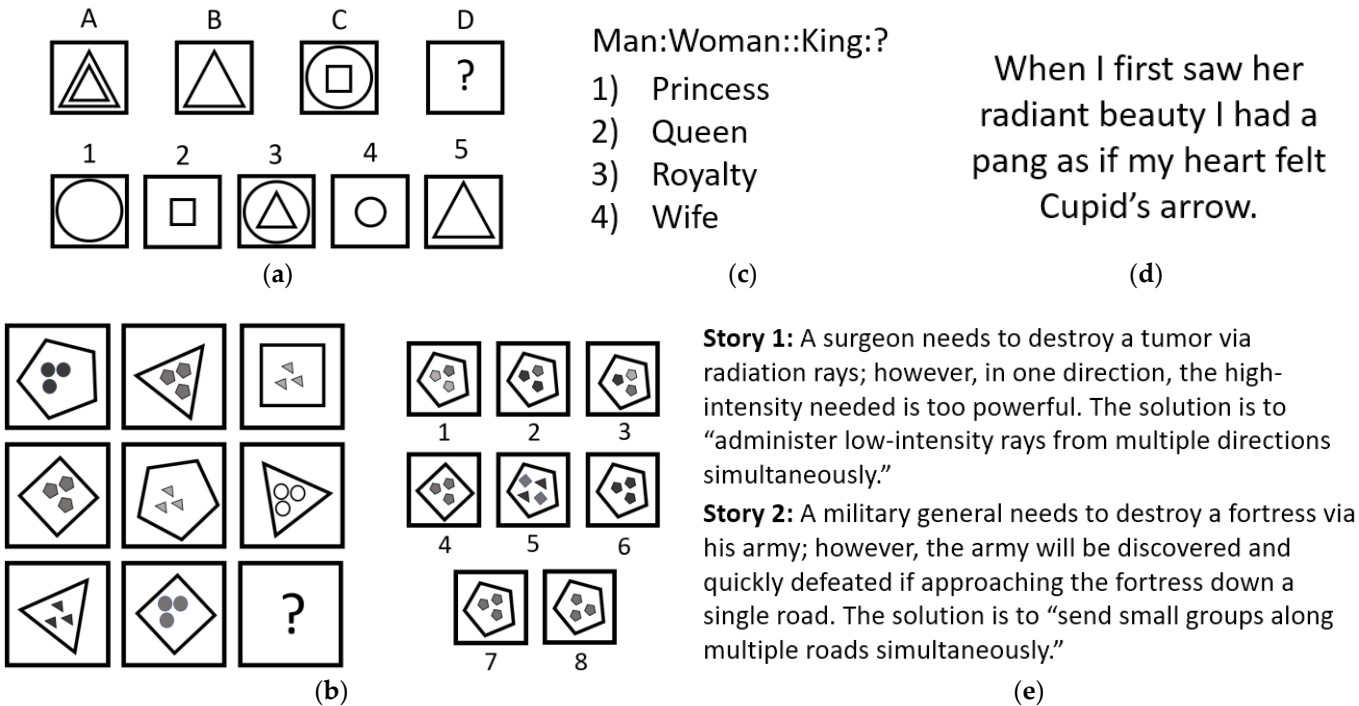

**Figure 6.** Example of visual and textual analogical reasoning problems; in the case of multiple choice questions (as in a,b,c), the numbers correspond to the answer choices (boxes with figures for a,b; word choices for c) presented to the viewer: (**a**) geometric A:B::C:D analogy problem utilized in Evans' 1964 ANALOGY algorithm (adapted with permission from Ref. [56]. 1964, Evans); (**b**) geometric matrix analogy problem from the RAVEN dataset (adapted with permission from Ref. [49]. 2019, Zhang et al.); (**c**) verbal word analogy problem in the multi-choice format of A:B::C:D (reprinted with permission from Ref. [63]. 2014, Levy & Goldberg.); (**d**) verbal sentence-based analogy problem (Reprinted with permission from Ref. [64]. 2001, Eliasmith.); (**e**) verbal story-based analogy problem (Adapted with permission from Ref. [65]. 1945, Duncker & Lee).

## 5. Comparisons of Transfer Learning and Analogical Inference in Two Application Domains

To determine the boundary between transfer learning models and analogical inference models, we take examples in the two most popular domains—computer vision and natural language processing—to compare algorithms, methods, and applications. The literature review presented here is brief, providing a light overview of both domains, rather than a comprehensive systematic literature review over these specific domains within transfer learning and analogical inference.

### 5.1. Computer Vision

Computer vision is concerned with allowing machines to perceive visual elements, objects, and scenes. Computer vision is one of the most popular applications of AI and machine learning domains, with the rise of self-driving cars and uncrewed aerial vehicles. Given the unfortunate potential for a small but critical perceptual error in the identification of a person or object by an autonomous system, it is exceptionally important that computer vision algorithms have high accuracy prior to their unsupervised introduction into the world. For a self-driving car, this may involve difficulty identifying a human crossing the street in a low-visibility situation. For military applications, an error could lead to an unintended attack on friendly forces if mistaken for hostile targets.

In general, many transfer learning applications in computer vision use a pre-trained model with a different classifier on the end, which is subject to fine-tuning for the new data at hand [66]. One example of this would be using an ANN already trained to rec-

ognize 1000 different classes (the source model) and adapting it so that the ANN is also trained to recognize other classes (resulting in the target model). Several transfer learning applications involve the same feature space but a different presentation of the data, such as facial recognition and expression [67–69]. Faces are similar based on their underlying structure and features; however, viewing a face at a different angle or comparing two faces expressing different emotions can be evaluated through transfer learning, as the target face appears slightly different than the source face. Specifically, in [67], the source dataset consisted of a stationary person, which was utilized in the target dataset involving a moving person. The application discussed in [68] involved face recognition, which varied based on the viewing angle, ethnicity, and general image conditions (such as lighting). A medical implementation of this type of transfer learning involves the evaluation of facial and body expressions for individuals with and without back pain [69]. Another active area of transfer learning involves object recognition in unfavorable conditions based on lighting [70,71], lack of general visibility [72,73], and/or image quality [74,75]. In these more typical autonomy-related computer vision applications, the clear benefit is the ability to determine the true label of an object without human input. The differences in viewing are often due to uncontrollable weather phenomenon, which self-driving cars and uncrewed aerial vehicles must be able to navigate successfully. Humans struggle with navigating in poor weather conditions due to distorted perception (rain, snow, etc.), which cannot be completely amended; thus, there is not a go-to human process to mimic, making this a difficult problem. Transfer learning has also been used to improve image quality [76] or generate brand-new images, called synthetic data (often through generative adversarial networks) [72]. The construction of synthetic data opens the doors for historically small datasets to be reasonably expanded without the need for combining datasets (if possible). This is beneficial because models typically show better performance with larger training, test, and/or validation datasets. For various reasons, some datasets are naturally limited to a small size, so by being able to artificially "expand" them through synthetic data, more accurate algorithms and models can be designed and implemented.

Over the past few decades, computational models of visual reasoning have been developed both in artificial intelligence [51,70,71] and cognitive science [5]. Despite many important differences, current models share certain core assumptions that are critical for their successes in accounting for human visual reasoning abilities but also lead to serious limitations. Fleuret et al. compared human and machine learning methods in a task that required categorizing images of novel shapes based on relational rules regarding the spatial arrangement of constituent parts [77]. The results showed that humans can efficiently acquire relational categories from a handful of exemplar images, but the performance of machine learning algorithms lagged far behind that of humans, even after exposure to thousands of labeled examples. In more recent work, deep learning models were trained using 1 million images generated according to relational rules [78]. For some reasoning problems, mainly involving spatial relations, the deep learning models reached human-level performance; however, for problems involving same–different comparative relations on certain visual attributes, the model performed much worse than humans, even after extensive training. Overall, computational accounts of visual reasoning using machine learning and deep learning approaches have so far fallen short of human-level performance for a wide range of visual reasoning problems, and (unlike humans) they rely on training with many similar problems.

In recent years, AI researchers developed visual analogy algorithms to solve visual reasoning problems, such as Raven's Progressive Matrices (RPM), first discussed in [47] and its extended versions in modern datasets, such as Procedurally Generated Matrices (PGM) [79], RAVEN [49], and Impartial-RAVEN [80]. These analogy problems are like the Bongard Problems originally discussed in [81] and later modernized into their namesake dataset [82]. Visual analogy problems and their real-world applications have been the subjects of a few recent review papers [6,83,84]. Though verbal and visual analogies are typically separate domains, there have been instances in which a verbal analogy algorithm

was re-utilized for visual problems [85,86]. Lu et al. applied the Bayesian Analogy with Relational Transformations (BART) model to an A:B::C:D dataset of individual objects with corresponding textual labels. Utilizing a multi-view learning scenario, BART was able to evaluate both the visual and semantic components to derive the correct solution for the given analogy.

Many models for solving RPM problems were created with inspiration from human studies, such as the Affine and Set Transformation Induction (ASTI) [87] and ASTI+ [88] algorithms. The idea behind many cognitive science applications is to match human performance, rather than achieve 100% accuracy, as is typically the goal in AI-based transfer learning applications. Posing a few-shot learning scenario using RPM problems, the meta-learning Analogical Contrastive Learning (meta-ACL) algorithm was developed to solve RPM by quantifying relational similarity between a source and target [89]. The meta-ACL algorithm was developed as a few-shot learning algorithm (subset of transfer learning) but has an emphasis on utilizing the relational structure of RPM problems to solve them [89]. This is one case where we can identify a marriage between transfer learning and analogical inference. The Rel-AIR (Relational Attend-Infer-Repeat) algorithm uses "unsupervised scene decomposition" to solve problems with the RAVEN and PGM datasets [90]. Rel-AIR splits elements of a given image into "segments," turns these segments into embeddings, and then reconstructs the image from ground zero until the original image is recreated. This process repeats such that the algorithm continues to learn about all the given information in RPM-like problems to derive what characteristics (such as size, color, and shape) are common.

One application that utilizes analogical mapping within zero-shot learning is visiPAM (visual Probabilistic Analogical Mapping), which takes labeled nodes on the images and infers the mapping between the two images [91]. Unlike the other previously mentioned algorithms, which focus on finding an answer (whether the answer must be completely generated or selected from a given list), visiPAM tackles the mapping problem. As mentioned earlier, compared to transfer learning techniques, analogical inference techniques have a greater emphasis on discovering the mapping or relational structure between the elements of two domains. Unlike RPM problems for which there is an inherent, deterministic answer, a mapping may differ based on the respondent, and there is not one distinct answer (as in the case of the problems visualized in Figure 5a,b).

In contrast to most current AI models, some models developed in cognitive science are capable of complex reasoning with a small amount of (or even no) training data. These models typically assume access to inputs with the expressive power of structured propositions. Some of the models use a symbolic code based on predicate calculus [57,92], whereas others employ neural codes that can capture role bindings, such as tensor products [93] or neural synchrony [94,95]. Structured representations make it possible to find systematic mappings across different contexts and to form relational generalizations. However, with a few important exceptions [82], most cognitive models simply hand-code the structured representations from images and have begged the basic question of where their structured representations come from. In practice, hand-coded representations restrict the applicability of models to relatively small "toy problems" (or at least to toy representations of what are much more complex problems). These models are unable to operate on the natural images that people encounter in their daily life. This limitation precludes taking advantage of databases that are now readily available as sources of "big data" for computational work. This difference in typical problem size between transfer learning and analogical inference techniques is one of the aspects distinguishing the two.

In summary, transfer learning applications in computer vision problems involve the utilization of feature learning or domain adaptation to "learn" elements within the training class and apply that to a similar, but new, domain for classification tasks. In contrast, analogical inference methods are focused more on representing relational structures in images and inferring the mapping between the source and the target.

*5.2. Natural Language Processing*

Natural language processing (NLP) focuses on allowing machines to produce human-like text and language. Natural language processing is a realm in which machines learn to process unstructured textual data in intelligent ways. Language inputs consist of pre-defined consistent building blocks—words—which can be tokenized and vectorized. Compared to computer vision applications, the consistent building blocks make applications of transfer learning and analogical inference in NLP more similar to one another. In general, text-based algorithms/models can be grouped into three categories: cognitive-science-inspired architectures, vector space models, and language transformer models (summarized in Table 3).

Major pioneering work on textual analogical inference includes the "structure mapping engine" (SME) [92]. SME is a symbolist algorithm, considered to be top-down in the sense an analogy is broken into its individual components (similar to Legos). Relations between elements (e.g., "a part of," "a type of") are developed to understand the underlying meaning of an analogy [92]. Many subsequent algorithms focus on connectionist (bottom-up approach) or hybrid approaches to map elements within a textual analogy problem (summarized in Figure 5c–e) [59]. Connectionist algorithms also break analogies into their individual components but place emphasis on the connectivity of these elements, not necessarily in specific relationships as symbolist algorithms. Hybrid algorithms combine elements of these two primary approaches (symbolist and connectionist).

More recently, vector space models, such as Word2Vec [96], Global Vectors (GloVe) [97], and FastText [98], have been applied to a wide range of semantics tasks, including verbal analogy (see reviews in [99–101]). Vector space models are a subset of connectionist algorithms, but unlike most cognitive-science-inspired algorithms, vector space models have been limited to four-term analogy problems, such as A:B::C:D-type analogies (e.g., Figure 6c), and have completely ignored the problem of mapping in analogy. Though vector space models can be trained, many are pre-trained on a large corpus of text from which word similarity is derived (often through cosine similarity).

Figure 7 shows a taxonomy of analogical inference "families" and their respective algorithms, presented previously in [59]. Most recently, language transformer models have been in the spotlight, including the Bidirectional Encoder Representations from Transformers (BERT) family [102–105], the Generative Pretrained Transfer (GPT) family of algorithms [106–108]), and XLNet [109] (see reviews in [110,111]). Language transformer models, especially within the GPT family, can perform realistic human-like language generation in addition to simple classification based on text inputs. Although these models are not explicitly trained to solve analogy problems, they have exhibited human-like performance in four-term analogy problems; vector space models and language transformer models are usually pre-trained on a corpus and are ready to be applied to language tasks, either within or outside of the pre-trained domain, making them frequently used for transfer learning [112]. The tasks for which analogical inference algorithms, vector space models, and transfer models are used primarily only overlap for *A:B::C:D* analogy problems, due to the limitations of vector space model. Language transformer models can be used in a wider variety of problems such as analogy/story generation.

One of the few applications of transfer learning that utilizes analogical inference maps similar, but different, domains with low-level representations [113], where "low-level" representations mean relatively basic relations. One application from transfer learning applies to story-based cross-domain analogies through Reproducing Kernel Hilbert Space. Utilizing first-order predicate logic and a vector space model (specifically Word2Vec), a symbol-based algorithm was used to extract relations in *A:B::C:D*-like analogies [114]. Although the algorithm is within the analogical inference domain, the usage of the Word2Vec algorithm also makes it an application of transfer learning. Most of the analogies discussed are semantic (emphasis on the meanings) rather than morphological (structure); however, multilingual morphological analogies are the focus of the Analogy Neural Network for the classification/retrieval (ANNc/ANNr) algorithm, which is compatible with, but does not

rely on, pre-trained model embeddings [115]. Within natural language processing, there is a question of how the model's performance changes depending on the task at hand, and of how distributed/localized the neurons in the algorithm's architecture are [116].

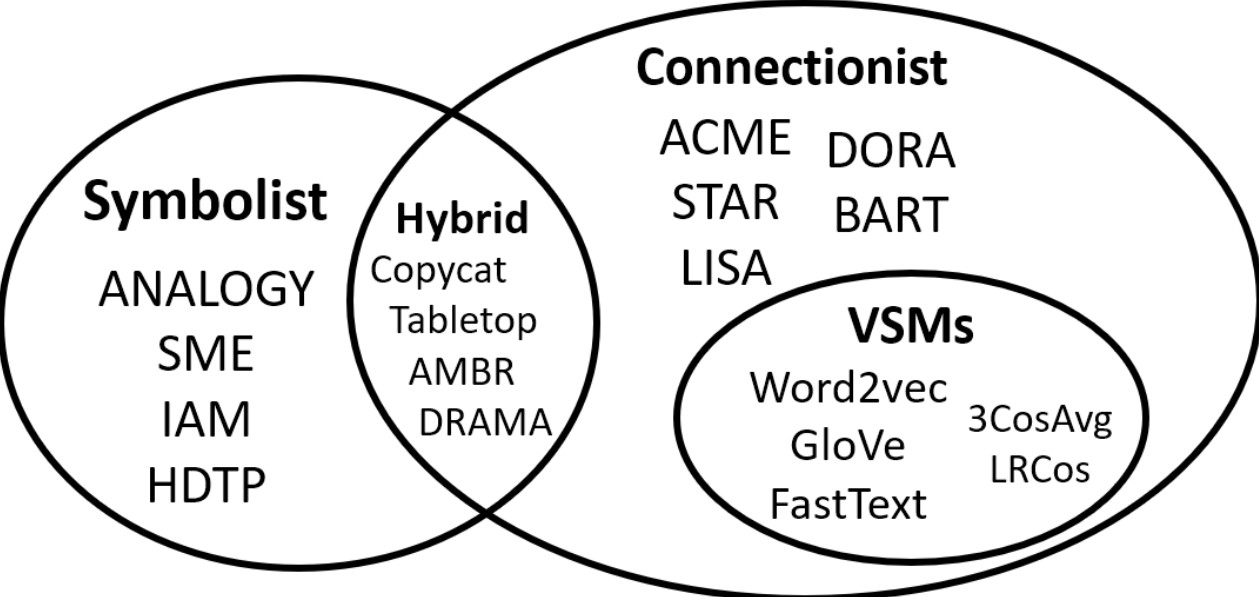

**Figure 7.** Taxonomy of symbolist, connectionist, and hybrid analogical inference algorithms (adapted with permission from Ref. [59]. 2022, Combs et al.).

The Bayesian Analogy with Relational Transformations (BART) family of algorithms (see [117,118]) solves A:B::C:D-like analogies by explicitly representing semantic relations between word pairs in a learned relation space. By extending relational representations to semantic networks using attributed graphs, recent work shows promising results in solving analogy problems using stories and a list of key words as inputs [119]. A single algorithm is not superior for verbal analogy tasks, nor are any of the three categories of models presented in Table 2. Analogical inference and transfer learning algorithms are implemented and applied very differently in natural language processing; however, both fields can benefit from leveraging developments from each other.

**Table 3.** Types of natural language processing algorithms.

| Category | Cognitive-Science-Inspired Architectures | Vector Space Models (VSMs) | Transformer Language Models |
|---|---|---|---|
| Primary task(s) | Map elements between a base domain to a target domain and/or infer new elements within the target domain | Derive quantitative relationships (typically similarity via the cosine distance) between two bodies of text via static word embeddings | Derive contextual meaning for completing general verbal tasks such as text generation, translation, etc. |
| Solvable Types of Analogy | | | |
| Word-based | Yes | Yes | Yes |
| Sentence-based | Yes * | No | Maybe |
| Story-based | Yes * | No | Maybe |

**Table 3.** *Cont.*

| Category | Cognitive-Science-Inspired Architectures | Vector Space Models (VSMs) | Transformer Language Models |
|---|---|---|---|
| Example algorithms/ models | <ul><li>Structure Mapping Engine (SME) [92]</li><li>Analogy Constraint</li><li>Mapping Engine (ACME) [120]</li><li>Associative</li><li>Memory-based</li><li>Reasoning (AMBR) [121]</li><li>Learning and Inferences with Schemas and Analogy (LISA) [94]</li><li>Distributed Representations for Analogy MApper (DRAMA) [64]</li><li>Discovery of Relations by</li><li>Analogy (DORA) [122]</li><li>Bayesian Analogy with Relational Transformations (BART) [117]</li></ul> | <ul><li>Word2Vec [96]</li><li>Global Vectors (GloVe) [97]</li><li>FastText [98]</li></ul> | <ul><li>Bidirectional Encoder Representations from Transformers family (BERT [102]; RoBERTa [103]; ALBERT [104]; DistilBERT [105])</li><li>Generative Pre-trained</li><li>Transformer family</li><li>(GPT [106]; GPT-2</li><li>[107]; GPT-3 [108])</li><li>XLNet [109]</li></ul> |

* BART focuses on relationship(s), so it is, therefore, only applicable to word-based analogies.

### 5.3. Summary

There are many similarities between transfer learning and analogical inference, but also several key differences, which are listed in Table 4. The characteristics of focus are transferred knowledge representation, tasks, amount of required training data, and the scope of transfer. Additional characteristics also differentiate applications of analogical inference and transfer learning; however, we believe that these are important to consider.

**Table 4.** High-level comparison between transfer learning and analogical inference.

| | Transfer Learning | Analogical Inference |
|---|---|---|
| Transferred Knowledge Representation | Feature spaces | Relational structures |
| Primary Tasks | Classification Generation | Classification Generation Mapping Retrieval |
| Amount of Required Training Data | Large amounts | Small amounts |
| Scope of Transfer | Near | Near and Far |

The emphasis on transferred knowledge representations is one of the key differences between transfer learning and analogical inference. Transfer learning is concerned with representing the source and target data by learning joint feature spaces (based on the results from feature extraction) for a particular type of task. A feature space is $n$-dimensional; however, for simplicity, a two-dimensional feature space utilizing two arbitrary features, X and Y, is shown in Figure 8. As mentioned earlier, a feature is best thought of as a numerical variable for the given data instance. Utilizing different shapes for the class of each data instance in Figure 8, the feature space can be divided such that there are predicted

"boundaries" for each data class, represented by different shapes. Hence, knowledge can be transferred in latent feature spaces shared between the source and target domains. In contrast, most algorithms in analogical inference focus on representations based on relational structure and the transfer of such structural knowledge between the source and target domains. Hence, capturing the similarity between relational structures in analogical inference prompts more abstract representations and thinking. A relational structure can be represented as a feature space; however, the feature space is not as essential to applications and methods in analogical inference when compared to transfer leaning. Transfer learning takes a more mathematical approach, involving the vectorization of data (often originally in image or text form) to develop a feature space for the given domain. These feature spaces may need to be adjusted when connecting or adjoining feature spaces for two domains to make them comparable. In transfer learning, any semantic information is represented in the data vectors; however, analogical inference may also represent data as a feature space. Analogical inference may also vectorize the data, but there is an emphasis on deriving semantic and relational meanings linking the two domains to make inferences.

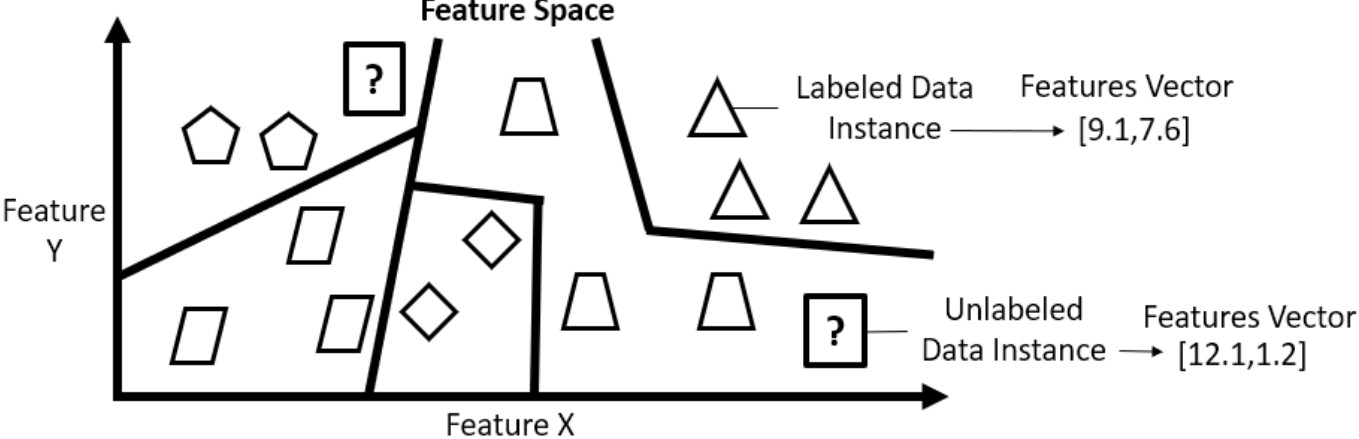

**Figure 8.** Example of a two-dimensional feature space.

Both transfer learning and analogical inference can be used for classification (placing in alike groups) and generation (creation of something new within the target domain); however, analogical inference can perform additional tasks, including mapping (drawing correspondences between the source and target domains) and retrieval (accessing short-term and long-term memory for a source domain given the target domain). Although retrieval, to some degree, occurs in all scenarios, analogical inference has more of a focus on what best-fitting instance should be retrieved. Retrieval for analogical inference is relatively flexible, whereas in transfer learning, the retrieval process is more deterministic. In some cases, these general tasks are carried out consecutively to fit the main task at hand. Specifically, in the computer vision realm, transfer learning applications predominantly are for image classification and generation, but analogical inference applications are concerned with solving analogy problems through the selection (more common) or generation (less common) of the "missing" element, as shown in Figure 6a,b. Within natural language processing, transfer learning is predominantly concerned with prediction tasks to generate human-like text and language, but semantic similarity between words is also extracted and can be used for solving some verbal analogy problems. Analogical inference is more concerned with solving a broader variety of analogy problems (encompassing sentence- and story-based analogies) in addition to word-based analogies. Analogical inference emphasizes forming representations for relations and concepts by minimizing the need for training with big data and discovering the similarity of relational structures across domains. Examples of some general and specific tasks are shown in Figure 9. Many of the exclusive transfer learning tasks and examples shown in Figure 9 concern utilizing either part of or all of a pre-trained model.

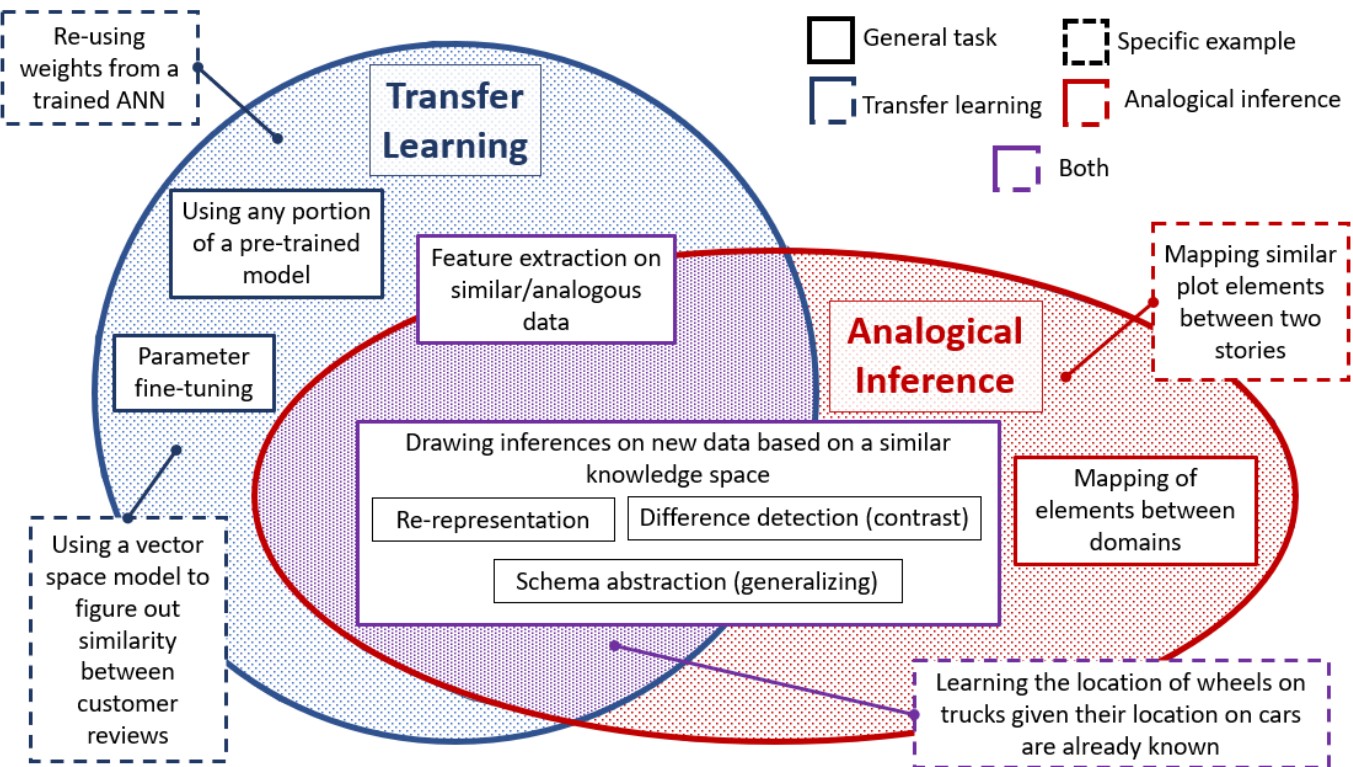

**Figure 9.** Venn diagram of separate and shared tasks and applications; blue denotes transfer learning tasks and examples, red denotes analogical inference tasks and examples, and purple denotes tasks and examples that may belong to either category.

Applications in the computer vision and natural language processing domains can be classified within the general tasks in Figure 9. The tasks discussed in Table 5 are examples; however, many more exist. It is important to note that transfer learning and analogical inference examples are not necessarily exclusive, and some applications can be both, depending on the algorithm and method(s) utilized.

Transfer learning almost always requires the source domain to have a large amount of high-quality labeled data, which are used to build a large, supervised model (such as ImageNet [123] for images or large text corpora for language). When classifications differ between the source and target domains, training data in target domains are necessary for transfer learning. For analogical inference, some knowledge is needed about the source domain, but the technique has also been successful with little or minimum training in the source domain (as in the case of Raven's Progressive Matrices problems [49]). In many problems related to analogical reasoning, humans do not receive training in the target domain and are unaware of what tasks may be performed in the target domain.

Finally, the scope of transfer concerns the information gap between the two domains. Knowledge about differences or similarities between the target and source domains pertains to their representations in feature spaces. Current transfer learning applications are centered on near transfer, which typically occurs when the source and target datasets are within the same general domain. For example, a source model that is able to classify dogs, cats, and horses might be expanded to also identify a target domain of bears and camels. This would likely be considered "near" transfer because all the items are animals with similar body structures (e.g., four legs). However, if the target domains were automobiles, trucks, and buses, the results would likely be very poor because of the dissimilarity of inherent feature spaces between the two domains. This would require "far" transfer, since an animal is different from a vehicle in many ways (e.g., four legs vs. wheels, material, articulated vs. rigid body) Analogical inference focuses on more abstract knowledge by explicitly taking into consideration between concepts so as to enable both near and far transfer. The

scope of transfer learning applications is often much more narrow than analogical inference applications; hence, transfer learning requires less focus on the mapping between a source and target domain. Returning to Figure 9, this is the primary explanation of why analogical inference goes beyond simply "reusing" pre-trained models and, instead, dives deeper into why and how two domains are similar.

**Table 5.** Specific examples of transfer learning and analogical inference task(s) within computer vision and natural language processing.

| Task Number | Task | Computer Vision | Natural Language Processing |
|---|---|---|---|
| 1 | Using any portion of a pre-trained model | Utilizing a pre-trained ANN architecture with frozen layers to predict a subset of the classes it was originally trained on | Utilizing a pre-trained vector space model to determine the similarity between two documents |
| 2 | Parameter fine-tuning | Utilizing a pre-trained ANN architecture with unfrozen layers to predict image classes in the target dataset<br>A more specific case of Task 1 | Adjusting the parameter values on a support vector machine to classify good and negative customer reviews |
| 3 | Feature extraction on similar/ analogous data | Transfer learning: Identifying important features on a target dataset given the features of a source dataset | Transfer learning: Identifying the most relevant words within a text document |
| | | Analogical inference: Given the important visual elements of the source portion of an analogy, identifying the important elements of its target counterpart | Analogical inference: Given the important textual elements of the source portion of an analogy, identifying the important elements of its target counterpart |
| 4 | Drawing inference on new data based on a similar knowledge space | Transfer learning: Identifying the location of four wheels on a truck given their locations on a car | Identifying the best D-word to complete the given incomplete textual analogy, *A:B::C:?* |
| | | Analogical inference: Inferring that since trucks have four wheels and cars are like trucks, cars must also have four wheels | |
| 5 | Mapping elements between two domains | Deciding which geometric elements of a source image corresponds best to the geometric elements of a target image (for example, utilizing *A* and *C* of Figure 6a, the large triangle of *A* can map to either the large or small square of *C*) | Given two analogous stories, identify which elements (characters, plot, setting, etc.) in the source story maps to the target story (for example, Figure 6e, determining whether to map the military general to the surgeon or the patient) |
| 6 | Re-representation | Identifying where a knee would be on a tree given its location on a human | Identifying the underlying common relationship between two different words that are usually not synonymous (for example, the relationship between the two sentences, "The runner trained for the marathon" and "The student studied for the test," could be re-representations of the word "preparing") |
| 8 | Schema abstraction (generalization) | Creation of a general representational structure (or schema) that derives from the identified relationships within the source and target data | |

## 6. Future Directions

We recommend that researchers carefully consider the differences between transfer learning and analogical inference before classifying research as one or the other, or perhaps

both. Although there are many elements that both techniques have in common, the differences we have reviewed show how the two approaches are distinctly different from one another. Future algorithms, methods, and applications should leverage concepts from both fields to advance the current state of AI. Transfer learning research still relies on significant amounts of pre-training within the source domain. The amount of data could be reduced through learning more abstract representations beyond feature spaces and learning an efficient mapping based on analogical structure. Research on transfer learning could benefit from considering how (dis)similar the source and target domains are, and not limiting the aims to near transfer, as has been typical.

Within analogical reasoning, utilizing current state-of-the-art advances in AI/ML in the transfer learning field has the potential to create more powerful and human-like algorithms that directly process raw inputs. Understanding human mechanisms for analogical reasoning and combining them appropriately with advances in transfer learning paves the way for biologically inspired processing. The technical aspect of transfer learning algorithms can lead to advances in cognitive science when evaluated and assessed from a human-based perspective. Many analogical inference applications utilize transfer learning already, with an emphasis on additional biologically inspired components to improve performance. However, it is important to understand that transfer learning and analogical inference applications may not always apply to the same situations in a given problem space. The important takeaway is that utilizing the algorithms, applications, and methods developed by the other discipline may foster both computational and conceptual advances.

## 7. Conclusions

Given the similarities between analogical inference and transfer learning, it is understandable that they are frequently treated as synonymous. However, a closer look at the underpinnings of each show field reveals that they are not necessarily mutually exclusive. Transfer learning is concerned with the re-utilization of a pre-trained model either in the same or different domains. In contrast, analogical inference, with its basis in cognitive science research, is more concerned with modeling the mechanisms by which the human brain draws inferences between a source and target scenario. Based on examples from computer vision and natural language processing, the two methods have different applications; however, there is also important overlap between the two. We hope our overview encourages mutual awareness and promotes synergy between the two fields.

**Author Contributions:** Conceptualization, K.C., H.L. and T.J.B.; Investigation, K.C.; visualization, K.C.; writing—original draft preparation, K.C.; writing—review and editing, H.L. and T.J.B. All authors have read and agreed to the published version of the manuscript.

**Funding:** This research was funded by the Air Force Research Laboratory through the Sensing, Learning, Autonomy, and Knowledge Engineering (SLAKE) contract, FA8650-19-C-1692.

**Data Availability Statement:** Not applicable.

**Acknowledgments:** The views expressed in this article are those of the authors and do not reflect any position or view of the United States Government, Department of Defense, or the Air Force. This work has been cleared for public release: distribution unlimited under AFRL-2023-1120.

**Conflicts of Interest:** The authors declare no conflict of interest.

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
