# Peer review of "Transfer Learning and Analogical Inference: A Critical Comparison of Algorithms, Methods, and Applications"

_algorithms, doi:10.3390/a16030146_

Round 1
Reviewer 1 Report
Quite an interesting article that touches on the difference between transfer learning and analogical inference, and how to integrate it. The study is also a unique review article, which presents methods of and background of machine learning, transfer learning and analogical inference. This research could be of interest to both the AI and cognitive science communities, as it addresses the understanding of both human and machine intelligence.
The article is well-written and interesting, and the subject matter is clearly relevant.
Some minor comments:
- The list of references is not entirely correct. Some references (e.g. 124-135) are not used in the text.
- Maybe the list of keywords should be expanded/revised (the title uses "analogical inference", and the list uses "analogical reasoning").
Author Response
Reviewer #1:
- The list of references is not entirely correct. Some references (e.g. 124-135) are not used in the text.
- Response: Unused references have been removed from the list.
- Maybe the list of keywords should be expanded/revised (the title uses "analogical inference", and the list uses "analogical reasoning").
- Response: For consistency, we have included “analogical inference” as a keyword. We use “analogical inference” throughout the paper to encompass multiple analogical-based processes (“analogical reasoning,” “analogical transfer,” “analogical mapping,” etc.); however, “analogical reasoning” is the most popular of these terms and is most used as a keyword in the literature.

Reviewer 2 Report
The paper presents a description and review of transfer learning and compare with analogical reasoning. Both techniques have some common uses and applications.
The paper is well written and describes both techniques without an in-deep review. It could be interesting as a first contact with both techniques, but it lacks a profound discussion. Application and algorithms of analogical inference are not well covered in the paper.
Some minor typos:
line 58: overlap > overlaps
line 322: quality [76]. or generate > quality [76] or generate
line 495: betwee > between
Author Response
Reviewer #2
- It could be interesting as a first contact with both techniques, but it lacks a profound discussion.
- Response: More article content, Figure 6 (pg. 18), and Table 5 (pg. 19) has been added in Section 5.3 to address this issue and create a better sense of discussion regarding how transfer learning and analogical inference are separate, but related fields. A more in-depth discussion can also be found in paragraph 1 of Section 6.
- Application and algorithms of analogical inference are not well covered in the paper.
- Response: Given the main intent is to draw parallels between transfer learning and analogical inference, the applications/algorithms section is included to provide a high-level review of some of the most prominent ones. This choice was made to ensure the paper was approachable due to length and readers unfamiliar with one or both of the disciplines. There are several thorough reviews of analogical inference algorithms referenced (such as [6, 83, 84, 110, 111]). However, some elaborations on the analogical inference algorithms mentioned were added throughout Section 5.
- Some minor typos:
- line 58: overlap > overlaps
- line 322: quality [76]. or generate > quality [76] or generate
- line 495: betwee > between
- Response: All noted instances above have been addressed in addition to others.

Reviewer 3 Report
In this paper, the authors review transfer learning algorithms, methods, and applications in comparison with work based on analogical inference to clarify differences and connections.
I can highlight the following issues of this paper:
1. Authors need to describe the major problem and aim of the paper in more details. Why need to compar two approaches?
2. Comparison is sketchy. Authors analyse well-known methods and algorithms. Need more details, examples, analytics, etc.
3. Results of the paper are very scoped. Need more detailed conclusions. How researchers can use analogical inference in popular data science tasks? How researchers can use two approaches together (if maybe)? How researchers can use some part of methods from analogical inference to improve transfer learning algorithms? Etc.
Author Response
Reviewer #3
- Authors need to describe the major problem and aim of the paper in more details. Why need to compare two approaches?
- Response: A few new sentences has been dedicated to this in the introductory section (Section 1); however, a more in-depth paragraph addressing this can be found in Section 2.
- Comparison is sketchy. Authors analyse well-known methods and algorithms. Need more details, examples, analytics, etc.
- Response: The newly added Figure 6 (pg. 18) and Table 5 (pg. 19) provides a more direct comparison between specific examples of tasks in each domain and their intersection. Section 5.3 has been significantly revised to reflect more details regarding the differences between transfer learning and analogical inference.
- Results of the paper are very scoped. Need more detailed conclusions. How researchers can use analogical inference in popular data science tasks? How researchers can use two approaches together (if maybe)? How researchers can use some part of methods from analogical inference to improve transfer learning algorithms? Etc.
- Response: This has been addressed in the first paragraph of Section 6 (pg. 20).

Round 2
Reviewer 3 Report
The paper can be accepted in the current form.